# ID-LoRA: Efficient Low-Rank Adaptation Inspired by Matrix Interpolative Decomposition

## Abstract

LoRA has become a universal Parameter-Efficient Fine-Tuning (PEFT) technique that equips Large Language Models (LLMs) to adapt quickly to new tasks. However, when these models are scaled up, even the latest LoRA variants still introduce considerable overhead in trainable parameters. Conversely, aggressively lowering the rank to curb this overhead markedly degrades performance in complex multi-task settings. We propose ID-LoRA, a novel PEFT framework that breaks the trade-off. Its core innovation lies in extracting and reusing clustered parameter groups from the pretrained weight matrix. These groups are then used to form multiple low-rank components, all of which share only a single initialized trainable low-rank matrix. This approach cuts the number of trainable parameters while keeping the model's capacity intact. We evaluate ID-LoRA on five diverse benchmarks: Mathematical Reasoning, Code Generation, MMLU, CommonsenseQA, and Safety Alignment. ID-LoRA outperforms both full fine-tuning and existing PEFT baselines (e.g., LoRA, DoRA, HydraLoRA) while using up to 46% fewer trainable parameters than the standard LoRA. In multi-task scenarios, it surpasses LoRA and its recent variants (e.g., DoRA and HydraLoRA) on both Code and MMLU tasks, yet requires only 54% of the trainable parameters demanded by the conventional LoRA.

## 1 Introduction

Large Language Models (LLMs) (Brown et al., 2020; Dubey et al., 2024; Yang et al., 2024) exhibit strong cross-task transfer capabilities, driving widespread adoption of a unified multi-task adaptation framework (Fang et al., 2024; Ding et al., 2023). In these frameworks, a single foundational LLM is fine-tuned to meet diverse downstream requirements. However, full-parameter fine-tuning is costly. Low Rank Adaptation (LoRA) (Hu et al., 2022) has become the standard choice because it is efficient. Yet it faces a critical trade-off: higher ranks linearly increase the number of trainable parameters and memory use, while a fixed low rank fails to adapt to multi-task scenarios.

To address this limitation, we propose ID-LoRA, a novel low-rank PEFT framework that fundamentally rethinks parameter efficiency. Inspired by Matrix Interpolative Decomposition (MID) (Ho & Greengard, 2012), ID-LoRA first clusters rows of the frozen pretrained weight matrix $W \in \mathbb{R}^{d \times d}$ (See Figure 1 (b)) into task-specific bases $\{A_i\}_{i=1}^{k}$ using k-means constrained with minimum cluster size (Levy-Kramer, 2018) with the Euclidean distance. These low-rank matrices $\{A_i\}_{i=1}^{k}$ are then frozen, and a shared trainable matrix $B$ is learned to compose the PEFT update jointly. During this process, we apply the Rank Boosting (RB) method to the intermediate inputs, further shrinking the size of the trainable matrix $B$ from $\mathbb{R}^{d \times r}$ to $\mathbb{R}^{\frac{d}{s} \times \frac{r}{s}}$.

The key innovation of ID-LoRA is to reuse clustered parameters from the pretrained weights as frozen auxiliary computations, instead of introducing a trainable matrix A like LoRA (See Figure 1 (a)). ID-LoRA trains only a shared matrix $B$ to compose bases via MID, enabling higher effective rank ($r$=128 in Figure 1 (c)) with lower trainable parameters. The superiority of ID-LoRA in multi-task scenarios stems from its theoretical guarantees. We prove in Theorem 2 that when $m$ tasks exhibit a $k$-cluster structure (Assumption 1), ID-LoRA's clustering-derived frozen basis matrices $\{A_i\}_{i=1}^{k}$ achieve tighter error bounds than the global low-rank adaptation, i.e., LoRA.

In experiments, we designed extensive single-task and multi-task scenarios. These experimental results collectively validate the effectiveness and efficiency of our method for multi-task adaptation. In single-task settings, ID-LoRA surpasses all baselines on mathematics, code generation, and safety tasks while updating only 0.56% of the parameters of LLaMA-3-8B. In multi-task experiments across MMLU benchmark, ID-LoRA reduces trainable parameters by 46% versus LoRA and yields a 6.0% absolute average gain over LoRA on LLaMA-3-8B. To broaden the evaluation diversity, we added three further benchmarks, namely Math, CommonsenseQA, and Code Generation, where ID-LoRA outperforms the baselines on two of them. The main contributions of our work are as follows: (1) We propose a PEFT method, ID-LoRA, which reuses frozen pretrained weights as low-rank bases and trains only a single shared matrix, eliminating the need for additional matrix $A$ parameters. (2) We prove that clustered decomposition yields tighter error bounds and improved pivot robustness, ensuring reliable performance in multi-task settings. (3) ID-LoRA achieves or surpasses LoRA-level accuracy while reducing trainable parameters by up to 46%.

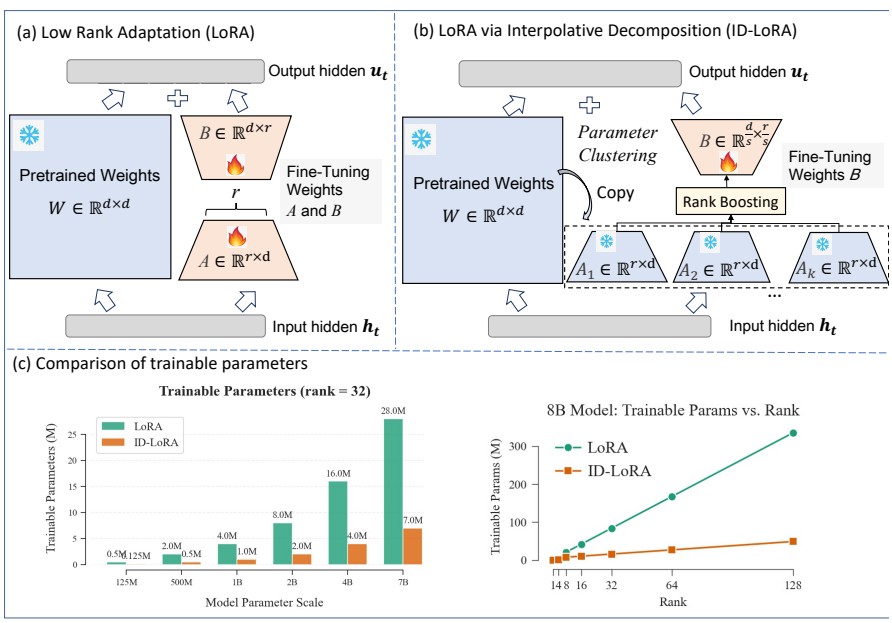

Figure 1: Architectural Comparison and Parameter Efficiency: LoRA versus ID-LoRA. (a) LoRA requires training two low-rank matrices: randomly initialized $A \in \mathbb{R}^{r \times d}$ and zero-initialized $B \in \mathbb{R}^{d \times r}$. (b) ID-LoRA employs the parameter clustering and rank boosting to generate multiple low-rank components while sharing a single B, thereby reducing trainable parameters. (c) Trainable parameters: ID-LoRA achieves $\sim 5\times$ compression versus LoRA at rank 32 (right) and maintains superior scalability across model sizes (left).

## 2 RELATED WORK

**Parameter-Efficient Fine-Tuning (PEFT).** PEFT aims to adapt LLMs to downstream tasks with minimal parameter changes. This line of research has led to a wide range of advancements (Houlsby et al., 2019; Hu et al., 2022; Huang et al., 2023; Tian et al., 2024). The three typical PEFT methods include **Adapter** (Houlsby et al., 2019), **Prefix-tuning** (Li & Liang, 2021) and **LoRA** (Hu et al., 2022). Adapter achieves adaptation to new tasks by inserting small adapter layers between the pre-trained layers, without retraining the entire model. This method has been used in various domains (Pfeiffer et al., 2021; Qiao et al., 2023). Prefix-tuning optimizes task-specific prefix vectors.

**LoRA** popularized low-rank decomposition for weight updates, balancing trainable parameters and performance. However, LoRA faces a critical trade-off: higher ranks improve task adaptation but linearly increase parameters and memory, while lower ranks degrade performance (Zhang et al., 2023; 2025). Several studies have proposed LoRA variants (Liu et al., 2024; Zhang et al., 2025; Kopiczko et al., 2024) to enhance learning capacity and reduce the number of trainable parameters.

Other studies(He et al., 2022) have provided a unified perspective, viewing many PEFT methods as a form of adapter. However, these techniques show constraints in multi-task scenarios.

**Multi-LoRA Adapters.** Researchers have explored the benefits of modeling multiple LoRA adapters for multi-task. LoraHub (Huang et al., 2023) adopts a multi-LoRA approach by training several adapters and selecting domain-specific combinations at inference. MoELoRA (Liu et al., 2023) methods leverages the Mixture-of-Experts (MoE) (Jacobs et al., 1991) to implement multi-domain knowledge modeling. HydraLoRA (Tian et al., 2024) uses an asymmetric structure to manage multiple LoRA adapters, and enhances parameter efficiency compared to existing symmetric approaches. These methods, however, necessitate training multiple adapters and consequently initializing additional trainable parameters.

Different from previous methods, the core innovation of ID-LoRA lies in constructing a frozen matrix by clustering the pretrained parameters, rather than initializing multiple independent LoRA adapters. This strategy significantly reduces the number of trainable parameters while maintaining adaptability to multi-task scenarios.

## 3 PRELIMINARIES

### 3.1 LOW RANK ADAPTATION

Low Rank Adaptation (LoRA) (Hu et al., 2022) introduces the concept of "intrinsic rank" in the parameter update process. This rank represents the true degrees of freedom needed for task adaptation. It is far smaller than the total parameter count, yet sufficient for effective fine-tuning. The insight is that pretrained language models have a minimal intrinsic dimension. Fine-tuning within this small subspace matches the effect of adjusting the entire parameter space (Aghajanyan et al., 2021). For the weight matrix $W$, its parameter update $\Delta W$ can be represented by low rank decomposition $A$ and $B$. In the fine-tuning phase, the adjustment is confined to low-rank matrices $A$ and $B$, while all other model parameters remain immutable. The update mechanism for these matrices is delineated in the subsequent equation:

$$u_t = Wh_t + \Delta W x = Wh_t + \frac{\alpha}{r}BAh_t \tag{1}$$

where $\alpha$ is the scaling factor, $W \in \mathbb{R}^{d \times d}$, $A \in \mathbb{R}^{r \times d}$ and $B \in \mathbb{R}^{d \times r}$, and $r < d$. To align with LoRA's initialization and ensure that $\Delta W$ is zero at the start of training, matrix $B$ in ID-LoRA is initialized to all zeros.

### 3.2 MATRIX INTERPOLATIVE DECOMPOSITION

Matrix Interpolative Decomposition (MID) (Horn & Johnson, 2012) is a structured low-rank matrix decomposition method: it identifies key rows (or columns) as a skeleton and approximates the original matrix via multiplication with a small low-rank matrix. In our work, we approximate $\Delta W$ by selectively reusing pretrained parameter matrix entries, reducing trainable parameters. For MID, we can write $\Delta W = BA$, where $A = W_{[S,:]}$ is formed by taking the rows of $W$ indexed by the set $S$, and $B$ is the low-rank coefficient matrix to be learn. In this way, a small number of rows of $W$ serve as the *skeleton* that approximates the matrix $\Delta W$. Specifically, the optimal row subset $S$ and the matrix $B$ are obtained by solving the following problem:

$$arg\,min||BW_{[S,:]} - \Delta W||_F \tag{2}$$

where $|S| = r$, and $\|\cdot\|_F$ is the matrix 2-norm. In MID, selecting the key rows from matrix $W$, namely $W_{[S,:]}$, is a non-trivial task. To mitigate this, we adopt a clustering strategy that generates several distinct row subsets, each acting as low-rank skeleton for a weighted a approximation of $\Delta W$.

## 4 METHOD

### 4.1 PARAMETER MATRIX ROW CLUSTERING

Given a parameter matrix $W \in \mathbb{R}^{d \times d}$, the rows of $W$ are treated as individual elements within a collection. These row vectors are partitioned into $k$ distinct clusters using the K-Means constrained with

minimum cluster size (Levy-Kramer, 2018) algorithm with the Euclidean distance metric. Within each cluster $C_i(i = 1, 2, ..., k)$, we select the $r$ row vectors exhibiting the smallest Euclidean distance to the cluster centroid $\mu_i$. The selected rows from each cluster $C_i$ are then aggregated to form a low-rank matrix $A_i \in \mathbb{R}^{r \times d}$. This process yields a set of $k$ structured low-rank matrices $A_i, A_2, \ldots, A_k$ that collectively capture the dominant row-space patterns of the original matrix $W$, prioritizing proximity to cluster centers. In our experiments, $k$ is equal to 4, and $r$ is set according to the specific experimental requirements.

To refine the approximation, we introduce an initialized trainable shared matrix $B \in \mathbb{R}^{d \times r}$ that reconstructs the updated parameter matrix $\Delta W$ jointly with each cluster-specific matrix $A_i$.

$$\Delta W = \sum_{i=1}^{k} \alpha_i (BA_i) \qquad (3)$$

where $\alpha_i$ is a scalar. As shown in Figure 2, $\alpha_i = T \odot A_i h_t$, where $\odot$ is the inner product. This blend retains key row patterns and efficiently tracks small updates to $\Delta W$.

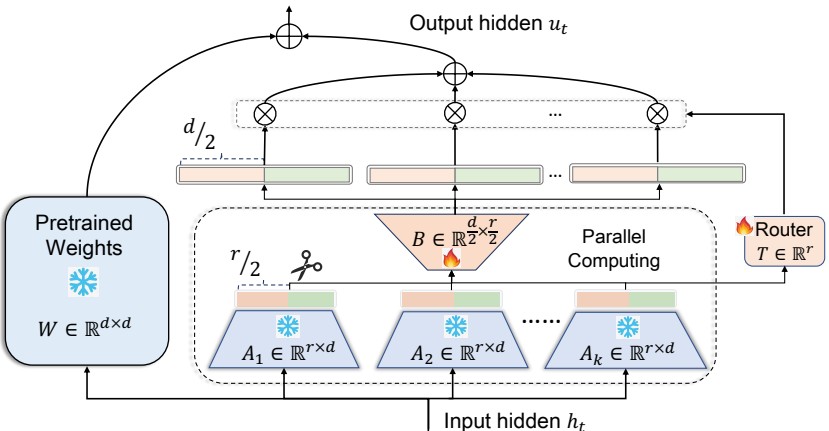

Figure 2: A diagram of the ID-LoRA architecture.

## 4.2 RANK BOOSTING

Relying on a single shared weight matrix $B$ significantly complicates achieving balanced and harmonious interactions among expert modules. To address this issue, we first partition the activation vector $x \in \mathbb{R}^r$ through matrix multiplication with $A_i$, then integrate these partitioned activation vectors via this unified shared low-rank matrix $B$, as shown in Figure 2. The formula is as follows:

$$x_i = A_i h_t, \ \ u_t = W h_t + \sum_{i=1}^{k} \alpha_i f_c([Bx_i^1, Bx_i^2]) \qquad (4)$$

where $B \in \mathbb{R}^{\frac{d}{2} \times \frac{r}{2}}$, $A_i \in \mathbb{R}^{r \times d}$, and $f_c$ is the concate operator. $x_i^1$ and $x_i^2$ are equal divisions from vector $x_i$. $\alpha_i$ is the output of router, i.e., $\alpha_i = T \odot x_i$, where $\odot$ is the inner product.

**Rank Analysis.** Compared with LoRA and its multi-task-oriented variant MoELoRA (Liu et al., 2023), our method simultaneously reduces the number of trainable parameters and preserves a higher rank. MoELoRA instantiates $k$ independent adapters $A_i \in \mathbb{R}^{r \times d}$ and $B_i \in \mathbb{R}^{d \times r}$, giving $k \times r \times (d + d)$ trainable parameters and, by the sub-additivity of matrix rank,

$$Rank(A_1 + A_2) \leq Rank(A_1) + Rank(A_2) \qquad (5)$$

For a single LoRA adapter, the number of trainable parameters is $r \times (d + d)$, and its rank is $r$. In contrast, we introduce a single low-rank matrix $B \in \mathbb{R}^{\frac{d}{2} \times \frac{r}{2}}$ and a lightweight router $T \in \mathbb{R}^r$. The parameter count is therefore only $(r \times d)/4 + r$. During the forward pass, the outputs of $k$ fixed matrices $\{A_1, A_2, \ldots, A_k\}$ are multiplied with $B$ and then concatenated, yielding a representation

whose rank is at most $k \times r$. Consequently, while employing significantly fewer trainable parameters, ID-LoRA attains a rank superior to that of both LoRA and MoELoRA.

## 4.3 THEORETICAL ANALYSIS OF CLUSTERED DECOMPOSITION

This section provides a concise theoretical justification for the two principal strengths of PMRC: (1) its ability to enhance multi-task learning via the extraction of multiple low-rank subspaces, and (2) its robustness to the choice of principal components.

First, we introduce three preliminary assumptions: (1) Task Clustering Structure: the optimal parameters of related tasks form disjoint groups; (2) Task Parameter Sharing: a single base matrix can be composed on-the-fly to yield task-specific parameters; (3) Pivot Sensitivity: the accuracy of standard low-rank factorizations, specifically the Column-Row-Update (CUR) decomposition as formalized by Penrose & Todd (1956), demonstrates dependence on the optimal selection of pivot indices during basis construction.

**Assumption 1 (Task Clustering Structure)** *Let $\{W_i^*\}_{i=1}^m \subset \mathbb{R}^{d_1 \times d_2}$ denote the optimal parameter matrices for $m$ related tasks. We assume these matrices can be partitioned into $k$ disjoint clusters $\mathcal{C}_1, \mathcal{C}_2 \ldots, \mathcal{C}_k$ (with $k < m$) such that:*

$$\forall i, j \in \mathcal{C}_l, \quad Rank(W_i^* - W_j^*) \leq r_l < \min(d_1, d_2) \tag{6}$$

*where $\mathcal{C}_l$ denotes the $l$-th cluster and $r_l$ is the intrinsic rank difference within the cluster.*

**Assumption 2 (Task Parameter Sharing)** *We assume the existence of a shared feature matrix $B \in \mathbb{R}^{d_2 \times r_g}$ with $r_g < d_2$, such that every optimal task-specific parameter matrix can be written as:*

$$W_i^* = BA_i + E_i, \quad \|E_i\|_F^2 \leq \epsilon \tag{7}$$

*where $A_i \in \mathbb{R}^{r_g \times d_1}$ is a task-specific and $E_i$ is the reconstruction error.*

**Definition 1 (Pivot Sensitivity)** *The CUR decomposition's approximation accuracy crucially depends on pivot selection, with reconstruction errors escalating sharply when the pivots inadequately capture the principal row or column subspaces,*

$$\max_{bad\ pivots} \|W - \tilde{W}_{CUR}\|_F^2 > \min_{good\ pivots} \|W - \tilde{W}_{CUR}\|_F^2 \tag{8}$$

Then, we proceed to formalize the guarantees of the PMRC through two theorems, Clustering Reconstruction and Cluster-Pivot Stability.

**Theorem 1 (Clustering Reconstruction)** *Under Assumptions 1 and 2, the clustering-aware decomposition achieves a tighter reconstruction error bound compared to global low rank decomposition:*

$$\sum_{i=1}^m \|W_i^* - \tilde{W}_i\|_F^2 \leq \sum_{i=1}^m \|W_i^* - \tilde{W}_i^{global}\|_F^2 - \Delta \tag{9}$$

*where $\Delta = \sum_{l=1}^k \sum_{i \in \mathcal{C}_l} \|B(A_{l(i)} - A^{global})\|_F^2 \geq 0$. The inequality becomes strict ($\Delta > 0$) when tasks exhibit clustering structure.*

**Theorem 2 (Cluster-Pivot Stability)** *Under Definition 1, the clustering-based local pivot selection ensures a tighter expected error bound compared to global pivot selection:*

$$\mathbb{E}_{\mathcal{P}}\left[\max_{1 \leq i \leq m} \|W_i^* - \tilde{W}_i^{\text{CUR}}\|_F^2\right] \geq \mathbb{E}_{\mathcal{P}_l}\left[\max_{1 \leq i \leq m} \|W_i^* - \tilde{W}_i\|_F^2\right], \tag{10}$$

*where $\tilde{W}_i^{\text{CUR}}$ is the reconstruction parameter using global pivots on all tasks, $\tilde{W}_i$ is reconstruction parameter using cluster-local pivots (per-cluster CUR on $\mathcal{C}_l$), and $\mathcal{P}_l$ is the set of pivots selected from cluster $\mathcal{C}_l$'s centroid matrix.*

Our theoretical framework is built on three mutually reinforcing assumptions that jointly yield two key guarantees. **Theorem 1** invokes Assumption 1 and Assumption 2 to establish that, whenever task clusters exist ($\Delta > 0$), cluster-aware decomposition strictly outperforms any global low-rank approximation. **Theorem 2** leverages Assumption 1 and Definition 1 to show that cluster-local pivot selection markedly reduces worst-case error sensitivity. Together, the results deliver both multi-task gains through structured decomposition and robustness to pivot choice, thereby establishing the method's soundness. Full proofs are provided in Appendix A.

| Method | # Params (%) | GSM8K | HumanEval | | | HEx-PHI |
| | | | Pass@1 | Pass@5 | Pass@10 | |
|---|---|---|---|---|---|---|
| *LLaMA-3-8B* | | | | | | |
| FFT | 100% | 58.8 | 30.5 | 39.3 | 41.7 | 94.8 |
| LoRA | 1.03% | 62.8 | 34.7 | 46.4 | 50.8 | 91.6 |
| DoRA | 1.05% | **63.5** | 33.1 | 44.0 | 48.6 | 93.6 |
| HydraLoRA | 1.10% | 62.8 | 33.4 | 44.7 | 49.3 | 94.2 |
| ID-LoRA(Ours) | 0.56% | **63.5**$^{\dagger}$ | **37.5** | **49.7** | **55.0** | **95.3**$^{\dagger}$ |
| *Mistral-7B* | | | | | | |
| FFT | 100% | 55.5 | 29.1 | 38.5 | 40.4 | 94.1 |
| LoRA | 1.15% | 57.8 | **33.8** | 42.4 | 45.3 | 91.9 |
| DoRA | 1.16% | 57.5 | 33.7 | 42.6 | 46.8 | 95.3 |
| HydraLoRA | 1.22% | **58.3** | 33.5 | 41.4 | 44.2 | 97.0 |
| ID-LoRA (Ours) | 0.62% | 58.1 | 33.5 | **43.3**$^{\dagger}$ | **46.9**$^{\dagger}$ | **97.2**$^{\dagger}$ |

Table 1: Performance comparison of different adaptation methods on single tasks, including GSM8K (Math), HumanEval (Code), and HEx-PHI (Safety) benchmarks using LLaMA-3 and Mistral. **Bold** indicates the best-performing method. † indicates statistically significant improvement, i.e., Wilcoxon signed-rank test ($p < 0.05$) against LoRA.

| Method | # Params (%) | MATH | MMLU | CQA | CODE | | |
| | | | | | Pass@1 | Pass@5 | Pass@10 |
|---|---|---|---|---|---|---|---|
| *LLaMA-3-8B* | | | | | | | |
| FFT | 100% | 52.2 | 37.5 | 38.7 | 31.2 | 40.1 | 43.9 |
| LoRA | 1.03% | **60.8** | 48.1 | 46.9 | 39.0 | 48.8 | 51.9 |
| DoRA | 1.05% | 59.4 | 41.5 | 35.0 | 38.3 | 50.2 | 55.7 |
| HydraLoRA | 1.10% | 60.1 | 47.5 | 44.1 | 33.2 | 43.9 | 47.8 |
| MoELoRA | 1.09% | 53.2 | 41.5 | 35.4 | 28.9 | 41.5 | 47.1 |
| ID-LoRA (Ours) | **0.56%** | 59.6 | **51.0** | **47.4** | **41.7** | **54.5** | **58.5** |
| *Mistral-7B* | | | | | | | |
| FFT | 100% | 41.9 | 46.7 | 62.6 | 23.0 | 30.2 | 33.9 |
| LoRA | 1.15% | **52.4** | 59.3 | 70.8 | 32.3 | 41.0 | 45.2 |
| DoRA | 1.16% | 51.8 | 59.4 | 71.1 | **32.5** | 40.1 | 43.3 |
| HydraLoRA | 1.22% | **52.4** | 60.4 | 71.3 | 32.4 | 41.7 | 45.6 |
| MoELoRA | 1.21% | 44.9 | 60.4 | 71.9 | 29.1 | 38.3 | 41.9 |
| ID-LoRA (Ours) | **0.62%** | 52.0 | **60.6**$^{\dagger}$ | **74.0** | 32.4 | **41.8**$^{\dagger}$ | **45.7**$^{\dagger}$ |

Table 2: Performance comparison of different adaptation methods on multi tasks, including math (GSM8K), code (HumanEval), world knowledge (MMLU), and common sense (CommonsenseQA) benchmarks using LLaMA-3 and Mistral. **Bold** indicates the best-performing method. "CQA" is the abbreviation for for CommonsenseQA. † indicates statistically significant improvement, i.e., Wilcoxon signed-rank test ($p < 0.05$) against LoRA.

## 5 EXPERIMENTS

To comprehensively evaluate ID-LoRA, we conduct experiments as follows: (1) Single-task experiments assessing general adaptation capability. (2) Multi-task experiments testing cross-domain transferability. (3) Efficiency analysis of time and memory. (4) Ablation studies on core components (i.e., PMRC and RB) and a performance comparison between LoRA and ID-LoRA under comparable trainable-parameter budgets. We conducted additional experiments to evaluate how the number of pretrained parameter clusters affects ID-LoRA's performance in multi-task scenarios. These results, along with other findings not presented in the main paper, are provided in Appendix B.3.

### 5.1 EXPERIMENTAL SETUP

We conduct systematic comparisons across both single-task and multi-task settings to holistically evaluate ID-LoRA's adaptability and scalability.

**Single-Task**. Three capabilities (Math, Code and Safety) are used to evaluate. (i) Mathematical Reasoning (**Math**): We fine-tune on GSM8K (Cobbe et al., 2021) training split and evaluated on the

test split. (ii) Code Generation (**Code**): We fine-tune on dataset CodeAlpaca (Chaudhary, 2023) and evaluated using pass@1, pass@5, and pass@10 on HumanEval (Chen et al., 2021). (iii) **Safety**: We fine-tune on Saferpaca (Bianchi et al., 2023), which extends Alpaca-Cleaned (Taori et al., 2023b) with 2,000 safety instructions. Safety performance is assessed by measuring the refusal rate on harmful queries from HEx-PHI (Qi et al., 2023).

**Multi-Task**. We conduct multi-task evaluation using the benchmark Multi-task Instruction Tuning: We jointly fine-tune on the concatenated training sets of Alpaca (Taori et al., 2023a), GSM8k (Cobbe et al., 2021) training split and 50% data of CodeAlpaca (Chaudhary, 2023). The mutli-task capability is evaluted on datasets including GSM8K test split (math), HumanEval (Chen et al., 2021) (code), MMLU (Hendrycks et al., 2021) (world knowledge) and CommonsenseQA (Talmor et al., 2019) (common sense).

**Baselines**. In single-task and multi-task experiments, we compare ID-LoRA with full fine-tuning (FFT), LoRA (Hu et al., 2022), DoRA (Liu et al., 2024) and HydraLoRA (Tian et al., 2024). In the multi-task setting, we expand the baseline pool established for single-task fine-tuning by additionally incorporating MoELoRA (Liu et al., 2023), a dedicated multi-task adaptation method.

**Model Configurations and Hardware**. All experiments were conducted on two NVIDIA A800 GPUs using LLaMA-3-8B (Grattafiori et al., 2024) and Mistral-7B (Jiang et al., 2023) as base models. To ensure a fair comparison, LoRA, DoRA, and HydraLoRA share identical hyper-parameter settings with ID-LoRA. Ablation studies additionally employ LLaMA-3.2-3B (AI, 2024). Complete hyperparameter details are provided in Appendix B.1.

## 5.2 EXPERIMENTS ON SINGLE-TASK

Table 1 demonstrates that ID-LoRA achieves superior parameter efficiency training only 0.56% of LLaMA-3-8B's parameters and 0.62% for Mistral-7B while consistently matching or outperforming all baselines. On the GSM8K mathematical reasoning task, ID-LoRA attains 63.5% with LLaMA-3, equaling DoRA's accuracy but with 50% fewer trainable parameters. For code generation (HumanEval), ID-LoRA delivers the highest Pass@10 scores (55.0% for LLaMA-3, 46.9% for Mistral), surpassing LoRA by 4.2–8.4% absolute gains. These results verife that ID-LoRA's low rank decomposition preserves critical task capabilities without compromising safety.

## 5.3 EXPERIMENTS ON MULTI-TASK

Table 2 shows that ID-LoRA outperforms LoRA and its variants across four domains while using only half the trainable parameters (0.56% on LLaMA-3 and 0.62% on Mistral). The key driver is the clustering-guided rank allocation: it concentrates updates on task-relevant subspaces, unlocking higher effective ranks without increasing the parameter budget. Consequently, ID-LoRA leads in five of six LLaMA-3 tasks and three of six Mistral tasks, with particularly large gains on knowledge-intensive benchmarks (e.g., MMLU and CommonsenseQA). On code generation it sets new Pass@10 records, whereas on math tasks it remains on par—suggesting that global weight updates are still crucial for mathematical reasoning.

The parameter configurations in Tables 1 and 2 reveal that while HydraLoRA employs a lower rank than other methods, it exhibits a larger number of trainable parameters. For instance, in Table 2 using Mistral-7B, HydraLoRA operates at a lower rank (r=16) compared to standard LoRA (r=32), yet requires more trainable parameters (1.22% versus 1.15%). This discrepancy stems from the parameter overhead introduced by HydraLoRA's asymmetric architecture. Conversely, the proposed ID-LoRA method achieves a higher rank (r=128) while maintaining only half the trainable parameters of standard LoRA. This dual advantage primarily originates from two key innovations: (1) constructing frozen matrices through clustering of pre-trained parameters, and (2) incorporating the RB algorithm.

## 5.4 EFFICIENCY ANALYSIS

Figure 3 shows the time and extra memory overhead during inference of different methods in multi-task experiments. During inference, since the adapters introduced by LoRA and DoRA can be merged with the original parameters, the model structure remains unchanged, avoiding additional

memory overhead and time latency. In contrast, the adapters introduced by MoELoRA and Hy-draLoRA can not be merged, resulting in extra memory usage and inference latency. The time consumption increased by 61.6% and 36.5% compared to LoRA. In our method, we only need to retain the clustering location information and the matrix $B$. ID-LoRA achieves a 45% reduction in extra memory usage relative to both MoELoRA and HydraLoRA, while incurring a minimal time overhead of only 0.5% compared to LoRA.

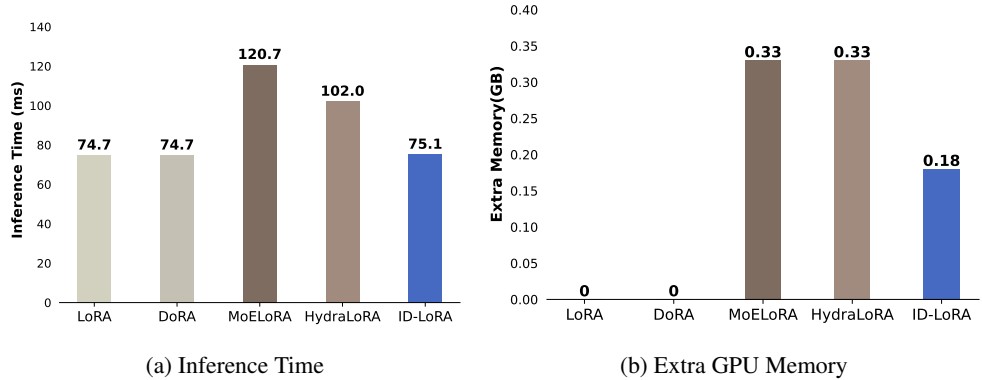

(a) Inference Time        (b) Extra GPU Memory

Figure 3: The inference time and extra memory overhead of different adaptation methods under the same hyperparameter settings as the multi-task experiments on LLaMA-3-8B, tested on an A800 GPU.

| Method | # Params (%) | MATH | MMLU | CQA | CODE Pass@1 | Pass@5 | Pass@10 |
|---|---|---|---|---|---|---|---|
| LoRA | 1.49% | **38.6** | 43.8 | 58.2 | 28.8 | 34.7 | 37.1 |
| ID-LoRA (PMRC+RB) | **0.77%** | 38.1 | **48.5** | **65.6** | **29.4** | **35.9** | **38.2** |
| ID-LoRA (w/o RB) | 0.83% | **38.8** | 45.2 | 64.4 | 28.7 | 35.3 | **38.6** |
| ID-LoRA (SS+RB) | 0.77% | 35.1 | 48.2 | 65.3 | 29.3 | 35.3 | 37.6 |
| ID-LoRA (RS+RB) | 0.77% | 36.6 | **48.5** | 65.1 | 28.9 | 34.5 | 37.4 |

Table 3: Ablation on LLaMA-3.2-3B across a multi-task dataset comparing different strategies. 'w/o RB' idenotes removing the RB algorithm from ID-LoRA. 'SS' abbreviates Serial Selection, where pivots are assigned to groups in sequential order. 'RS' abbreviates Random Selection, where pivots are assigned to groups at random.

## 5.5 ABLATION ANALYSIS

We first conducted a parameter-parity performance comparison between LoRA and ID-LoRA. Ablations then dissect the contribution of each design component: (i) PMRC Effectiveness: Comparing clustered row selection against random and serial selection strategies. (ii )RB Advantage: Evaluating rank scalability and multi-task performance under matched trainable parameter budgets.

### 5.5.1 PARAMETER-PARITY PERFORMANCE ANALYSIS

Figure 4 illustrates the performance of ID-LoRA versus LoRA across four benchmarks in a multi-task setting. When both methods operate under approximately equal trainable-parameter budgets, ID-LoRA consistently surpasses LoRA, and the margin grows as the parameter ratio increases, underscoring its superior parameter efficiency and robustness. These findings confirm that, under comparable parameter constraints, ID-LoRA can employ larger ranks, leading to additional gains in multi-task modeling.

### 5.5.2 ABLATION EXPERIMENTS ON PMRC AND RB

To validate PMRC's clustering-based grouping, we compare it with two baselines on LLaMA-3.2-3B under a multi-task setup: Serial Selection (SS) takes contiguous rows and Random Selection (RS) samples rows uniformly from the frozen pretrained weight matrix. Table 3 contrasts

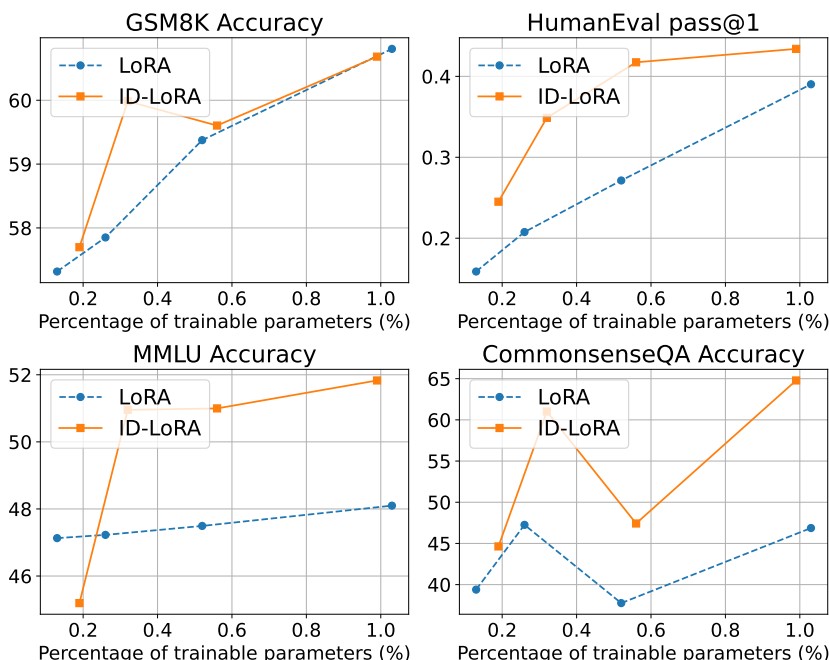

Figure 4: Performance comparison between ID-LoRA and vanilla LoRA on four representative benchmarks (GSM8K, HumanEval, MMLU, and CommonsenseQA) using LLaMA3-8B as the backbone. Both methods vary the rank while keeping the number of trainable parameters approximately equal, and results are reported under few-shot or zero-shot settings. Detailed results are provided in Appendix B.2

PMRC-based grouping with two baselines, Serial (SS+RB) and Random (RS+RB) row selection, on LLaMA-3.2-3B. With equal parameter budgets, PMRC consistently outperforms both baselines across all tasks, confirming that clustering-based initialization enables superior expert specialization and downstream gains.

Table 3 presents a controlled ablation on the LLaMA-3.2-3B multi-task benchmark to isolate the contribution of RB (PMRC+RB vs w/o RB). To ensure parity, we configure ID-LoRA with rank $r = 128$ closely matching the parameter budget of the variant that omits RB. Across four multi-task evaluation benchmarks, the RB-enhanced variant delivers consistent gains, except on MATH. MATH tasks necessitate precise and accurate directional updates to the weight matrix; simply increasing the rank through the RB algorithm alone is insufficient to enhance performance. These results validate the intrinsic effectiveness of the RB algorithm in multi-task scenarios.

## 6 CONCLUSION AND FUTURE WORK

In this paper, we propose ID-LoRA, a PEFT framework that integrates matrix interpolation decomposition with clustered low-rank adaptation. Theoretically, we prove that cluster-aware decomposition strictly improves upon global low-rank approximations when task clusters exist. Empirically, we show that ID-LoRA not only improves performance in single-task, but also maintains strong results across multiple multi-task benchmarks while reducing trainable parameters to 46% below those of standard LoRA.

Although ID-LoRA achieves strong results, its gains are less pronounced on mathematical-reasoning benchmarks than on other tasks, where it remains largely on par with LoRA and its recent variants. This limitation arises because mathematical reasoning demands precise directional weight updates rather than mere rank expansion. Building on these findings, future work will focus on integrating the ID-LoRA architecture to develop even more efficient learning strategies that further enhance model performance.

STATEMENT

**Ethics statement.** This paper proposes a novel efficient fine-tuning architecture that improves upon LoRA-based frameworks. It enhances modeling capability in multi-task scenarios while reducing the number of trainable parameters. We do not identify any potential negative concerns.

**Reproducibility statement.** This paper provides all necessary technique details for reproducibility, including theoretical analysis, algorithm details, experimental settings, and source code of the proposed techniques.

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

# A  THEORETICAL FRAMEWORK OF ID-LORA

To preserve theoretical completeness, we first restate the three assumptions from the paper, then, derive Theorems 1 and 2 via the detailed definitions that follow, providing full proofs in this section.

## A.1  PROBLEM SETUP AND ASSUMPTIONS

**Assumption 3 (Task Clustering Structure)** *Let $\{W_i^*\}_{i=1}^m \subset \mathbb{R}^{d_1 \times d_2}$ denote the optimal parameter matrices for $m$ related tasks. We assume these matrices can be partitioned into $k$ disjoint clusters $\mathcal{C}_1, \mathcal{C}_2 \ldots, \mathcal{C}_k$ (with $k < m$) such that:*

$$\forall i, j \in \mathcal{C}_l, \quad Rank(W_i^* - W_j^*) \leq r_l < \min(d_1, d_2) \tag{11}$$

*where $\mathcal{C}_l$ denotes the $l$-th cluster, and $r_l$ is the intrinsic rank difference within the cluster.*

**Assumption 4 (Task Parameter Sharing)** *We assume the existence of a shared feature matrix $B \in \mathbb{R}^{d_2 \times r_g}$ with $r_g < d_2$, such that every optimal task-specific parameter matrix can be written as:*

$$W_i^* = BA_i + E_i, \quad \|E_i\|_F^2 \leq \epsilon \tag{12}$$

*where $A_i \in \mathbb{R}^{r_g \times d_1}$ is a task-specific matrix and $E_i$ is the reconstruction error.*

**Definition 2 (Pivot Sensitivity)** *The CUR decomposition's approximation accuracy crucially depends on pivot selection, with reconstruction errors escalating sharply when the pivots inadequately capture the principal row or column subspaces,*

$$\max_{bad\ pivots} \|W - \tilde{W}_{\text{CUR}}\|_F^2 > \min_{good\ pivots} \|W - \tilde{W}_{\text{CUR}}\|_F^2 \tag{13}$$

## A.2  CORE DEFINITIONS

**Definition 3 (Task Parameter Distance)** *For any pair of tasks $i, j \in [m]$, their distance is defined as the squared Frobenius norm of the difference between their optimal parameter matrices:*

$$d(i, j) = \|W_i^* - W_j^*\|_F^2 \tag{14}$$

**Definition 4 (Cluster-aware Low-Rank Decomposition)** *Let the optimal parameter matrices of the $m$ tasks be $\{W_i^*\}_{i=1}^m \in \mathbb{R}^{d_1 \times d_2}$.*

*(1) Perform $k$-means on $\{W_i^*\}_{i=1}^m$ to obtain disjoint clusters $\{\mathcal{C}_l\}_{l=1}^k$.*

*(2) For each cluster $\mathcal{C}_l$, compute the centroid matrix $M_l = \frac{1}{|\mathcal{C}_l|} \sum_{i \in \mathcal{C}_l} W_i^*$ and extract its rank $r_l$ truncated SVD, yielding the cluster-specific matrix $A_l \in \mathbb{R}^{d_1 \times r_l}$.*

*(3) Learn the shared matrix $B \in \mathbb{R}^{d_2 \times r}$ by solving*

$$\min_B \sum_{i=1}^m \|W_i^* - BA_{l(i)}\|_F^2, \tag{15}$$

*where $l(i)$ denotes the cluster of task $i$.*

**Definition 5 (Multi-Task Reconstruction)** *The parameter matrix for task $i$ is reconstructed as*

$$\tilde{W}_i = BA_{l(i)} \tag{16}$$

**Definition 6 (Column-Row Update (CUR) Decomposition)** *Given a matrix $W \in \mathbb{R}^{d \times d}$, a CUR decomposition approximates $W$ as a product of three matrices:*

$$W \approx CUR \tag{17}$$

*where $C \in \mathbb{R}^{d \times c}$, and is a subset of $c$ columns from $W$. $R \in \mathbb{R}^{r \times n}$ and is a subset of $r$ rows from $W$. $U \in \mathbb{R}^{c \times r}$, which is a weight matrix constructed to minimize the approximation error $\|A - CUR\|_F$.*

A.3   MAIN THEOREMS

**Theorem 3 (Clustering Reconstruction)** *Under Assumptions 1 and 2, the clustering-aware decomposition achieves a tighter reconstruction error bound compared to global low rank decomposition:*

$$\sum_{i=1}^{m} \|W_i^* - \tilde{W}_i\|_F^2 \le \sum_{i=1}^{m} \|W_i^* - \tilde{W}_i^{global}\|_F^2 - \Delta \tag{18}$$

*where $\Delta = \sum_{l=1}^{k} \sum_{i \in \mathcal{C}_l} \|B(A_{l(i)} - A^{global})\|_F^2 \ge 0$. The inequality becomes strict ($\Delta > 0$) when tasks exhibit clustering structure.*

**Proof.** Let $\tilde{W}_i^{\text{global}} = BA^{\text{global}}$ denote the reconstruction obtained from the globally shared low-rank factorization. By **Assumption 2**, we have

$$\left\|W_i^* - \tilde{W}_i^{\text{global}}\right\|_F^2 = \left\|B(A_i - A^{\text{global}}) + E_i\right\|_F^2. \tag{19}$$

In addition, the cluster-aware reconstruction (i.e., **Definition 2**) satisfies

$$\left\|W_i^* - \tilde{W}_i\right\|_F^2 = \left\|B(A_i - A_{l(i)}) + E_i\right\|_F^2. \tag{20}$$

Since $A_{l(i)}$ is the cluster centroid, we have

$$\|A_i - A_{l(i)}\|_F^2 \le \|A_i - A^{\text{global}}\|_F^2, \tag{21}$$

and $B$ is optimized to minimize the within-cluster approximation error. Consequently, $\Delta \ge 0$. Whenever the inter-cluster task discrepancy is significant, $\Delta > 0$, demonstrating the superiority of the cluster-aware decomposition.

**Theorem 4 (Cluster-Pivot Stability)** *Under Definition 1, the clustering-based local pivot selection ensures a tighter expected error bound compared to global pivot selection:*

$$\mathbb{E}_{\mathcal{P}} \left[ \max_{1 \le i \le m} \|W_i^* - \tilde{W}_i^{\text{CUR}}\|_F^2 \right] \ge \mathbb{E}_{\mathcal{P}_l} \left[ \max_{1 \le i \le m} \|W_i^* - \tilde{W}_i\|_F^2 \right], \tag{22}$$

*where $\tilde{W}_i^{\text{CUR}}$ is the reconstruction parameter using global pivots on all tasks, $\tilde{W}_i$ is the reconstruction parameter using cluster-local pivots (per-cluster CUR on $\mathcal{C}_l$), and $\mathcal{P}_l$ is the set of pivots selected from cluster $\mathcal{C}_l$'s centroid matrix.*

**Proof.** Step 1: Error Decomposition and Notation.

(1) *Global Approximation Error*: For task $i$, the global CUR approximation is:

$$\tilde{W}_i^{\text{CUR}} = C_{\mathcal{P}} U_{\mathcal{P}} R_{\mathcal{P}}, \quad \mathcal{P} \subseteq [d_1] \times [d_2], |\mathcal{P}| = s \tag{23}$$

where pivot set $\mathcal{P}$ is uniformly sampled from all rows or columns.

(2) *Local Approximation Error*: For task $i$ in cluster $\mathcal{C}_l$:

$$\tilde{W}_i = C_{\mathcal{P}_l} U_{\mathcal{P}_l} R_{\mathcal{P}_l}, \mathcal{P}_l \sim \text{cluster-specific distribution} \tag{24}$$

with pivots $\mathcal{P}_l$ selected from cluster-specific rows or columns.

Step 2: Lower Bound for Global Error.

By Definition 1, there exists at least one cluster $\mathcal{C}_{l^*}$.

(1) Rare Feature Non-Ignorability:

$$\mathbb{P}_{\mathcal{P}}(\mathbf{v}_{l^*} \notin \text{span}(C_{\mathcal{P}})) \ge \eta_{l^*} > 0 \tag{25}$$

where $\mathbf{v}_{l^*}$ is the unique feature direction of $\mathcal{C}_l$, $\|\mathbf{v}_{l^*}\|_2 = 1$. Since it has negligible weight in the global feature space (e.g., $\sigma_{l^*} < \sum \sigma_i$).

(2) Error Lower Bound: when $\mathbf{v}_{l^*}$ is missed:

$$\|W_i^* - \tilde{W}_i^{\text{CUR}}\|_F \ge \gamma_{l^*} \quad \forall i \in \mathcal{C}_{l^*} \tag{26}$$

Thus:

$$\mathbb{E}_{\mathcal{P}}\left[\max_i \|W_i^* - \tilde{W}_i^{\text{CUR}}\|_F\right] \geq \gamma_{l*} \cdot \eta_{l*} \tag{27}$$

where $\gamma_{l*} = \sup_{\mathcal{P}:\mathbf{v}_{l*} \notin \text{span}(C_{\mathcal{P}})} \|W_i^* - \tilde{W}_i^{\text{CUR}}\|_F$ (minimum error when $\mathbf{v}_{l*}$ is missed globally).

Step 3: Upper Bound for Local Error.

For any cluster $\mathcal{C}_l$, local pivot selection satisfies:

(1) Feature Coverage Property:

$$\mathbb{P}_{\mathcal{P}_l}(\mathbf{v}_l \in \text{span}(C_{\mathcal{P}_l})) \geq 1 - \epsilon_l \quad (\epsilon_l \leq e^{-c|\mathcal{P}_l|}) \tag{28}$$

because $\mathbf{v}_l$ is prominent within $\mathcal{C}_l$ (by Assumption 1).

(2) Error Decomposition:

$$\|W_i^* - \tilde{W}_i\|_F \leq \underbrace{\|M_l - \tilde{M}_l\|_F}_{\text{cluster center error}} + \underbrace{\|W_i^* - M_l\|_F}_{\leq \delta_l} \tag{29}$$

where $\tilde{M}_l = C_{\mathcal{P}_l} U_{\mathcal{P}_l} R_{\mathcal{P}_l}$.

(3) Low-Rank Approximation: By Assumption 1 ($\text{rank}(M_l) \leq r_l$) and CUR theory:

$$\|M_l - \tilde{M}_l\|_F \leq (1 + \sqrt{r_l})\|M_l - M_l^{(r_l)}\|_F = 0 \tag{30}$$

when $\mathbf{v}_l \in \text{span}(C_{\mathcal{P}_l})$. Hence:

$$\mathcal{E}_l^{\text{local}} \leq \delta_l \quad \text{with probability} \geq 1 - \epsilon_l \tag{31}$$

where $\delta_l = \max_{i \in \mathcal{C}_l} \|W_i^* - M_l\|_F$ (intra-cluster consistency measure).

(4) Expectation Control:

$$\mathbb{E}[\mathcal{E}_l^{\text{local}}] \leq \delta_l(1 - \epsilon_l) + (\text{diam}(\mathcal{C}_l) + \|M_l\|_F)\epsilon_l \leq \delta_l + \mathcal{O}(\epsilon_l) \tag{32}$$

where $\text{diam}(\mathcal{C}_l) = \max_{i,j \in \mathcal{C}_l} \|W_i^* - W_j^*\|_F$.

Step 4: Global vs. Local Error Comparison:

$$\mathbb{E}[\mathcal{E}_{\text{global}}] \geq \gamma_{l*}\eta_{l*}, \quad \mathbb{E}[\mathcal{E}_{\text{local}}] \leq \max_l(\delta_l + \mathcal{O}(\epsilon_l)) \tag{33}$$

From Definition 1 ($\gamma_{l*}\eta_{l*} > \delta_l + \epsilon_l$), the theorem follows. First, global error lower bound depends on rare feature miss probability $\eta_{l*}$ and minimum miss error $\gamma_{l*}$. Then, local error upper bound is controlled by intra-cluster consistency $\delta_l$ and feature coverage failure $\epsilon_l$.

| Hyperparameter | GSM8K | | CodeAlpaca | | Saferpaca | |
|---|---|---|---|---|---|---|
| | Llama-3 | Mistral | Llama-3 | Mistral | Llama-3 | Mistral |
| Rank/$\alpha$ | 128 / 256 | 128 / 256 | 128 / 256 | 128 / 256 | 128 / 256 | 128 / 256 |
| Learning Rate | 5e-5 | 3e-5 | 5e-5 | 5e-5 | 1e-4 | 1e-4 |
| Dropout | 0.05 | 0.05 | 0.05 | 0.05 | 0.05 | 0.05 |
| Optimizer | AdamW | AdamW | AdamW | AdamW | AdamW | AdamW |
| Batch Size | 32 | 32 | 32 | 32 | 32 | 32 |
| Warmup Steps | 0 | 0 | 0 | 0 | 0 | 0 |
| Epochs | 3 | 3 | 2 | 2 | 2 | 2 |

Table 4: Hyperparameter settings for ID-LoRA on GSM8K, CodeAlpaca, and Saferpaca datasets. The ID-LoRA parameters are placed at the q, k, v, o, gate, up, and down positions.

# B EXPERIMENTAL DETAILS AND ADDITIONAL RESULTS

This appendix provides comprehensive experimental details omitted from the main text due to space constraints, along with additional results that further validate the effectiveness of our method. We first restate all hyper-parameter settings, and computational-resource specifications to ensure full

| Method | ID-LoRA | ID-LoRA |
|---|---|---|
| Base Model | Llama-3 | Mistral |
| Rank $r$ | 128 | 128 |
| $\alpha$ | 256 | 256 |
| Learning Rate | 5e-5 | 5e-5 |
| Dropout | | 0.05 |
| Optimizer | | AdamW |
| Batch size | | 32 |
| Warmup Steps | | 0 |
| Epochs | | 2 |
| Where | | q, k, v, o, gate, up, down |

Table 5: Hyperparameter settings for ID-LoRA on the multi-task dataset

| | Rank | LoRA # parameters | DoRA # parameters | HydraLoRA # parameters | MoELoRA # parameters | ID-LoRA # parameters |
|---|---|---|---|---|---|---|
| LLAMA3-8B | 8 | 21.0M | 22.3M | 46.7M | 88.9M | 7.7M |
| | 16 | 41.9M | 43.3M | 89.7M | 172.8M | 10.4M |
| | 32 | 83.9M | 85.3M | 175.7M | 340.5M | 16.0M |
| | 64 | 167.8M | 169.1M | 347.7M | 676.1M | 27.0M |
| | 128 | 335.5M | 336.9M | 691.6M | 1.3B | 49.0M |

Table 6: Number of trainable parameters for different methods on LLaMA-3-8B

reproducibility. We then conduct extended ablation studies on key design choices, examining how the selection of the cluster number $k$ influences multi-task performance, and we include additional multi-task evaluations that were omitted from the main paper. Table 6 presents the number of trainable parameters under identical rank settings for different methods on the LLaMA 3-8B model. These results show that ID-LoRA requires the fewest trainable parameters among baseline methods under the same rank.

## B.1 HYPER PARAMETER SETTINGS

In this section, we provide detailed hyperparameter settings for the main experiments in the paper, which are presented in Tables 4 and 5. Table 4 details the fine-tuning configurations of the ID-LoRA method in single-task settings for Math, Code, and Safety tasks, applied to two base models: LLaMA-3-8B and Mistral-7B. Table 5 show the hyperparameter settings for multi-task scenarios.

| Method | # Params (%) | MATH | MMLU | CQA | CODE | | |
|---|---|---|---|---|---|---|---|
| | | | | | Pass@1 | Pass@5 | Pass@10 |
| LoRA | 0.13 (r=4) | 57.3 | 47.1 | 39.4 | 15.9 | 24.6 | 29.1 |
| | 0.26 (r=8) | 57.9 | 47.2 | 47.3 | 20.7 | 30.3 | 34.5 |
| | 0.52 (r=16) | 59.4 | 47.5 | 37.8 | 27.1 | 37.3 | 42.2 |
| | 1.03 (r=32) | 60.8 | 48.1 | 46.9 | 39.0 | 48.8 | 51.9 |
| ID-LoRA | 0.19 (r=32) | 57.7 | 45.2 | 44.6 | 24.5 | 32.7 | 36.2 |
| | 0.32 (r=64) | 60.0 | 51.0 | 61.0 | 34.9 | 47.7 | 53.7 |
| | 0.56 (r=128) | 59.6 | 51.0 | 47.4 | 41.7 | 54.5 | 58.5 |
| | 0.99 (r=256) | 60.7 | 51.8 | 64.8 | 43.4 | 55.9 | 59.9 |

Table 7: Performance comparison of LoRA and ID-LoRA on the LLaMA-3-8B backbone under multi-task instruction tuning. All methods are trained on the same mixture (Alpaca + GSM8k + 50% CodeAlpaca) and evaluated on MATH, MMLU, CommonsenseQA (CQA), and HumanEval (Pass@1/5/10). These numbers correspond to the experiment shown in Figure 4 of the paper.

## B.2 EXPERIMENTAL DETAILS OF PARAMETER-PARITY PERFORMANCE

The results in Table 7 provide the detailed numerical data for Figure 4, which could not be fully presented in the main text due to space constraints. As the proportion of trainable parameters rises from 0.13% to 1%, both methods improve on every task. ID-LoRA already surpasses LoRA at 0.56% parameters, attaining 59.6 on MATH and 41.7 Pass@1 on HumanEval, and achieves the

overall best performance at 0.99% parameters (MATH 60.7, MMLU 51.8, HumanEval Pass@1 43.4).

## B.3  ABLATION: IMPACT OF PRETRAINED PARAMETER CLUSTER NUMBER $k$

| $k$ | MATH | MMLU | COM | CODE | | |
| --- | --- | --- | --- | --- | --- | --- |
| | | | | Pass@1 | Pass@5 | Pass@10 |
| 2 | 38.3 | **48.7** | 64.4 | 29.0 | 34.7 | 37.1 |
| 4 | 38.1 | 48.5 | **65.6** | 29.4 | 35.9 | 38.2 |
| 6 | **39.6** | 46.9 | 65.1 | **30.6** | **36.5** | **38.4** |
| 8 | 38.1 | 45.5 | 64.7 | 30.2 | 36.0 | 38.2 |

Table 8: Ablation on the number of pre-trained parameter clusters k under the multi-task setting. Results are obtained with the LLaMA-3.2-3B backbone. The table reports overall multi-task accuracy for each choice of $k$.

To ablate the effect of the cluster count on model behavior, we conducted multi-task experiments on LLaMA-3.2-3B with ranks fixed at r=128 and cluster numbers set to $k = 2, 4, 6, 8$. The outcomes, reported in Table 8, guided our choice of $k = 4$ (or $k = 6$) for the main experiments.

As the cluster number $k$ increases, the amount of copied parameters in the pre-trained parameter matrix also rises accordingly, which reduces the distinction between different sub-matrices $A_i$, weakening the diverse combinatorial advantages brought by the routing mechanism. The performance decline at $k = 8$ reflects this issue of diminished expressive power caused by excessive replication.

