# OpenReview forum: "ID-LoRA: Efficient Low-Rank Adaptation Inspired by Matrix Interpolative Decomposition"
_ICLR.cc/2026/Conference — Submitted to ICLR 2026_

### Official Review · Reviewer_jCmp · 2025-10-28

**Soundness:** 3
**Presentation:** 3
**Contribution:** 3
**Rating:** 6
**Confidence:** 3

**Summary:**

ID-LoRA is a novel parameter-efficient fine-tuning (PEFT) method for LLMs that reduces trainable parameters by reusing clustered weight groups to form multiple low-rank components with a single shared trainable matrix. It outperforms full fine-tuning and existing PEFT methods (LoRA, DoRA, HydraLoRA) across diverse benchmarks—including reasoning, code generation, and multi-task settings—while using up to 46% fewer trainable parameters.

**Strengths:**

1. ID-LoRA reuses frozen pretrained weights as low-rank bases and trains only a single shared matrix, removing extra parameters.
2. Clustered decomposition provides tighter error bounds and enhanced pivot robustness for reliable multi-task performance.
3. Achieves LoRA-level or better accuracy while reducing trainable parameters by up to 46%.

**Weaknesses:**

1. ID-LoRA introduces some computational and memory overhead, which may compromise the inference-time advantages of standard LoRA.
2. The authors could provide a comparison of fine-tuning costs between ID-LoRA and other LoRA variants. While it uses fewer trainable parameters, this does not necessarily imply reduced training cost or improved training efficiency.
3. The authors could provide more details on how ID-LoRA operates during inference which would be helpful.

**Questions:**

Please refer to the Weaknesses section.

---

> ### Author Response · Authors · 2025-11-20
>
> We sincerely appreciate your thoughtful and constructive comments. We have carefully considered each of your questions and provide detailed responses below.
>
> ### **W1 Computational and memory overhead**
>
> ID-LoRA reuses rows of the original weight matrix $W$ for its $A_i$ matrices. Although it introduces new parameters B and a router layer, these add only a negligible number of parameters relative to LoRA and therefore do not materially increase GPU-memory consumption.
> Moreover, the corresponding computations overlap with the original forward pass, so the impact on inference latency is minimal. As already reported in Figure 3 of our paper, on LLaMA-3-8B the extra delay versus LoRA (with full weight merging) is merely ~0.5 % and the memory overhead is only 1.1 %, yet ID-LoRA still outperforms other efficient fine-tuning methods in both memory footprint and inference time.
>
>
> ### **W2 Fine-tuning cost**
>
> As you suggested, we have conducted training time measures (in hours) on a single A800 GPU, using the same batch size (i.e., b_s = 32) and the same number of training steps (i.e., train_step = 1000). The results are shown in the table below:
>
> | lora | dora | hydralora |  moelora | idlora |
> | :---: | :---: | :---: | :---: | :---: |
> | *LLaMA-3-8B* | | | | | | |
> | 1.03 | 1.81 | 1.48 | 1.99 | **1.27** |
>
> The results indicate that training cost follows a nearly identical trend to inference cost: while it is slightly higher than LoRA's, it remains lower than other efficient fine-tuning methods.
>
>
> ### **W3 Details during inference**
> Thank you for this suggestion. We will supplement the detailed operational flow of ID-LoRA at the inference stage: during inference, we first compute
> $$x = Wh_t$$
>
> Then, using the indices obtained during model-parameter clustering, namely $S_1, S_2, ..., S_i,...,S_k$, we perform the corresponding index selection on $x$:
> $$x_i = s(x, S_i)$$
> where $i\in[1,2,...,k]$, and $s(\cdot)$ is the value-selection function that picks entries from the vector from $x$ according to the index set $S_i$.
> Next, we compute Eq. (4) in the paper:
> $$ u_t = x+\sum_{i=1}^{k} \alpha_i f_c([Bx_i^1, Bx_i^2]) $$
> where $f_c(\cdot)$ is the concatenation function.
>
> If you have any further comment, please feel free to let us know and we are more than glad to discuss with you.

---

### Official Review · Reviewer_7pCV · 2025-10-31

**Soundness:** 2
**Presentation:** 3
**Contribution:** 2
**Rating:** 4
**Confidence:** 3

**Summary:**

This paper presents a PEFT framework, ID-LoRA, that cleverly reuses parts of the pretrained model's weights to reduce the number of trainable parameters. The core idea of clustering frozen weights to form task-specific bases is well-motivated and is a significant departure from conventional LoRA-based methods (see Figure 1). The empirical results are strong, demonstrating superior parameter efficiency and performance across a diverse set of single-task and multi-task benchmarks (see Tables 1 & 2). However, the experimental evaluation, while broad, lacks depth in several areas, such as hyperparameter sensitivity, the impact of specific design choices, and analysis on larger-scale models. Furthermore, the clarity of certain methodological details and the connection between the theoretical analysis and practical implementation could be improved (see Section 4.2, Section 4.3).

**Strengths:**

1. **Novel Parameter Efficiency**
   The core idea of reusing clustered pretrained parameters as frozen bases is innovative. It reduces trainable parameters by up to 46% compared to standard LoRA while maintaining competitive performance.

2. **Solid Theoretical Grounding**
   The paper provides rigorous theoretical guarantees (Theorems 1-2) showing tighter error bounds for cluster-based decomposition compared to global low-rank adaptation.

3. **Comprehensive Multi-Task Evaluation**
   Extensive experiments across 5 diverse benchmarks (Math, Code, MMLU, CommonsenseQA, Safety) demonstrate consistent advantages over LoRA variants in multi-task scenarios.

4. **Practical Inference Efficiency**
   ID-LoRA achieves minimal inference overhead (only 0.5% latency increase) compared to methods like MoELoRA (+61.6%) and HydraLoRA (+36.5%), while reducing extra memory usage by 45%.

**Weaknesses:**

1. **Ambiguity in Mathematical Formulation**

The description of the Rank Boosting (RB) mechanism is unclear, and the associated formula exhibits dimensional inconsistencies. Evidence from Section 4.2, Equation (4) shows the term `fc([x1_i, x2_i]B)`, where `x1_i` and `x2_i` are partitions of a vector of size `r`, resulting in a concatenated vector of size `r`. However, matrix `B` is defined as having dimensions `R^(r/2) x (d/2)`, making standard matrix-vector multiplication impossible. This ambiguity hinders understanding and reproducibility of the core method, undermining its practicality.

2. **Limited Experimental Scope and Rigor**

The experiments, while broad, lack depth in validation: model scale is confined to 7B/8B parameters without extension to larger models (e.g., 8B+), failing to demonstrate parameter efficiency in critical scenarios; results are reported as single-run scores without error bars or confidence intervals from multiple runs, raising doubts about statistical significance; training time analysis is omitted, neglecting quantification of k-means clustering overhead; clustering choices lack justification, with no comparison to alternative algorithms (e.g., cosine similarity) or validation of distance metrics; hyperparameters like minimum cluster size are unspecified, and pivot selection strategies (e.g., centroid-based) are not ablated against alternatives; ablation on cluster count `k` is only on 3B models with marginal differences, insufficient for robust conclusions; RB motivation is vaguely described (e.g., "balanced interactions"), lacking technical explanation. These limitations reduce the robustness and persuasiveness of the experimental findings.

3. **Disconnect Between Theory and Practice**

The theoretical analysis, based on CUR decomposition and pivot selection (e.g., Assumption 3 and Theorem 2), is loosely connected to the method's implementation, which uses k-means clustering and centroid-distance-based row selection. Evidence indicates that Theorem 2 discusses "cluster-local pivots" stability, but the algorithm does not select pivots in a CUR-typical manner, making the theoretical guarantees less directly applicable. This gap weakens the claim that the method's success stems from its theoretical foundations.

4. **Missing Qualitative Analysis**

The paper lacks qualitative insights into the method's behavior, such as analyzing whether clusters correspond to specific linguistic or reasoning capabilities, or how the router parameters α activate clusters for different tasks or inputs. This omission limits opportunities to understand internal mechanisms beyond quantitative metrics, reducing the depth of methodological validation.

**Questions:**

1. **Rank Boosting Scaling Factor**
   The RB method currently uses a division factor of 2. Could it theoretically be extended to /4 or even extreme cases where B becomes d×1 for greater parameter reduction? Ablation studies should explore the impact of different scaling factors on performance to determine the optimal value.

2. **Router Mechanism Analysis**
   The router is essentially a learnable alpha parameter. Could fixed alpha values be used? Statistical analysis of alpha values across experiments should be conducted—what are their average values? Are there observable patterns?

3. **Mergeability and Inference Latency**
   Due to the alpha-based router, can the adapter be merged with pretrained weights during inference? If not, does this significantly increase inference latency?

4. **Comparison with LoRA-FA**
   LoRA-FA [5] also trains only B while keeping A fixed (Gaussian initialization). How does ID-LoRA compare comprehensively with LoRA-FA to further validate PMRC's effectiveness?

5. **Baseline Completeness**
   This work aims to reduce trainable parameters through sharing, but relevant baselines are missing (e.g., VeRA [1], Tied-LoRA [2], BS-LoRA [3], VB-LoRA [4]). Related work should be discussed, and experiments should include comparisons with other parameter-sharing methods.

6. **Figure 1(c) Misleading Representation**
   The x-axis in Figure 1(c) increases non-linearly, exaggerating parameter savings exponentially. For fair comparison, ranks should grow linearly—the current plot overstates ID-LoRA’s gains. Since savings are fixed (~1/8), a linear plot may be more appropriate.

7. **Diagram Consistency**
   Figure 1 uses K=3 clusters, but experiments use K=4. Shouldn’t these be consistent?

---

> ### Author Response · Authors · 2025-11-21
>
> Thank you for your recognition of the innovation, theoretical foundation, and experimental evaluation of our ID-LoRA.
> We have carefully considered each of your questions and provide detailed responses below.
>
> **W1: Regarding the Misunderstanding of Equation (4)**
>
> The term $fc([x^1_i, x^2_i]B)$ in Equation (4) means that vector $x^1_i$ and vector $x^2_i$ are multiplied with matrix $B$ separately, and then concatenated using the concat function $fc$. To avoid misunderstanding, we have adjusted the term to $fc([Bx^1_i, Bx^2_i])$. The revised version of the paper has been updated.
>
> **W2: Misunderstandings and Clarifications Regarding the Experiments**
>
> 1. The model scale was indeed limited to 3B/7B/8B parameters at consumer-grade level, as experiments with larger models were constrained by computational resources;
>
> 2. Our results are not from single-run scores. In our experiments (i.e., Table 1 and Table 2), most results are averages from multiple runs, and the statistical significance tests have been annotated.
>
> 3. We conducted multi-task training experiments on a single A800 GPU, using the same batch size (i.e., b_s = 32) and the same number of training steps (i.e., train_step = 1000). We then recorded the training time (in hours) for each method to assess the corresponding training cost. The experimental results are shown in the table below:
>
> | lora | dora | hydralora |  moelora | idlora |
> | :---: | :---: | :---: | :---: | :---: |
> | *LLaMA-3-8B* | | | | | | |
> | 1.03 | 1.81 | 1.48 | 1.99 | **1.27** |
>
> The experimental results show that the training-cost trend closely parallels the inference-cost trend: compared with LoRA it increases only slightly, and it remains lower than that of other efficient fine-tuning methods.
>
> 4. The computational overhead of k-means clustering has been experimentally evaluated on the new Qwen3-8B model, with additional analysis included as follows: 'Using the k-means-constrained clustering algorithm, clustering 7 linear layers in one decoder layer takes approximately 8 minutes on a single-core Intel(R) Xeon(R) CPU E5-2683 v3 processor. Among these, the most complex layers (up_proj or gate_proj) require about 2.5 minutes each for clustering. Since clustering of different layers can be performed concurrently, the total time can be reduced to as low as 2.5 minutes in a multi-process scenario. Further optimization through parallel clustering computation for individual linear layers on GPUs could potentially reduce this time even more.
>
> 5. ID-LoRA does not focus on the clustering method itself. Clustering is merely a technique we selected to construct frozen parameter groups.
> Regarding the parameter of the minimum number of clusters $k$.
> We conducted ablation experiments with k = 2, 4, 6, and 8, as shown in Table 8. Based on the performance results in the table, we selected k=4 as the hyperparameter for our main experiments.
>
> 6. The motivation of RB is to increase the modeling capacity of the rank while maintaining a comparable number of trainable parameters. We will add this description to the introduction in the next version.
>
> **W3 Explanation of the Theoretical Logic**
>
> The stability and quality of low-rank approximations depend heavily on the selection of "pivots".
> We introduce a new method that uses a clustering-based approach to choose these pivots. In this method, multiple cluster centers correspond to multiple tasks.
> Our theoretical analysis shows that this clustering method effectively achieves the intended benefits.
> Although it works differently from traditional pivot selection algorithms, it solves the same core problem: ensuring high-quality and stable pivot selection in a multi-task setting.
>
> **W4 Parameters $\alpha$ Qualitative Analysis**
>
> The answer is in **Q2**.

---

> > ### Author Response · Authors · 2025-11-21
> >
> > **Q1 Rank Boosting Scaling Factor**
> >
> > Although theoretically feasible, it cannot be further subdivided in practice.
> > This is due to the fixed dimensions of the parameter matrices in the base model, and the limitation on the number of vectors in each cluster, i.e., the number of rows in the A_i matrix.
> >
> > On our selected model scales (3B/8B), the RB method does not allow intermediate activations to be split into more segments. For the corresponding number of clusters (k=4), splitting into 2 segments proves to be the optimal choice.
> >
> > **Q2 Router Mechanism Analysis**
> >
> > We conducted the following experiment. We sampled 100 examples from each of the code, math, and cqa test datasets, and collect the ID-LoRA router activation values from the *self_attn.q_proj* layer at the 20th decoder layer during the computation of the first token in the reasoning process.
> > The results (i.e., Mean and Standard Deviation (Std Dev)) are shown in the table below.
> >
> > *(1) Statistical results of all tasks from the Routers*
> > | Statistics | Router 0 | Router 1 | Router 2 | Router 3 |
> > |:-:|:-:|:-:|:-:|:-:|
> > | **Mean** | 0.2026 | 0.1777 | 0.1968 | 0.4220 |
> > | **Std Dev** | 0.0924 | 0.0350 | 0.0439 | 0.1097 |
> >
> > *(2) The results ofRouters Statistical are by Task Type*
> >
> > Mean
> > | Task | Router 0 | Router 1 | Router 2 | Router 3 |
> > |-|:-:|:-:|:-:|:-:|
> > |**code**|  0.2646 | 0.1606 | 0.2181 | 0.3566 |
> > |**math**|  0.1019 | 0.2049 | 0.1585 | 0.5347 |
> > |**cqa**|  0.3197 | 0.1469 | 0.2485 | 0.2849 |
> >
> >
> > Std Dev
> > | Task | Router 0 | Router 1 | Router 2 | Router 3 |
> > |:-:|:-:|:-:|:-:|:-:|
> > |**code**| 0.0325 | 0.0245 | 0.0360 | 0.0490 |
> > |**math**| 0.0142 | 0.0287 | 0.0170 | 0.0476 |
> > |**cqa**| 0.0204 | 0.0155 | 0.0180 | 0.0252 |
> >
> > Based on the observed in router activation values across different task types during inference, a fixed $\alpha$ should not be used.
> >
> > **Q3 Mergeability and Inference Latency**
> >
> > The adapter consists of two parts: $\{A_i\}$ and B. Part $\{A_i\}$ originates directly from the original pre-trained parameters, and does not require merging.
> > Part B cannot be merged.
> >
> > ID-LoRA does not significantly increase inference latency.
> > The experimental result is shown on the Figre 3 (a) of this paper.
> > Compared to the LoRA method that merges the adapter back into the pretrained weights, ID-LoRA introduces only a marginal increase in inference time.
> >
> > **Q4 Comparison with LoRA-FA**
> >
> > Although LoRA-FA also trains $B$ while keeping $A$ frozen, the advantage of ID-LoRA over LoRA-FA is that it employs multiple sets of $A$, and through a Router mechanism, it can better adapt to multi-task scenarios.
> > The multiple sets of $A$ are constructed using the PMRC method.
> >
> > **Q5 Baseline Completeness**
> >
> > We have included VeRA (Note that methods like Tied-LoRA, BS-LoRA, and VB-LoRA are similar to VeRA) as  additional baseline for a comparative evaluation with ID-LoRA under a comparable trainable parameter budget. The experimental results On LLaMA3-8B are as follows:
> >
> > |Method|MATH|MMLU|CQA|CODE Pass@1|CODE Pass@5|CODE Pass@10|
> > |:---:|:---:|:---:|:---:|:---:|:---:|:---:|
> > |VeRA 	|55.9|	46.2|	40.7|	34.6|	46.4|	50.3|
> > |ID-LoRA|59.6|	51.0|	47.4|	41.7|	54.5|	58.5|
> >
> > These experimental results demonstrate the advantages of ID-LoRA.
> >
> > **Q6 Figure 1(c) Misleading Representation**
> >
> > The x-axis in the right panel of Figure 1(c) has been corrected, and the plot has been replaced with a bar chart for comparison.
> >
> > **Q7 Diagram Consistency**
> >
> > The problem has been consistent in the new version of the paper.

---

> > > ### Comment · Reviewer_7pCV · 2025-11-26
> > >
> > > Thank you for your detailed response and for adding the VeRA baseline as well as the router analysis. I have read your rebuttal carefully. While the additional experiments address some of my concerns regarding baselines, I remain unconvinced about the fundamental soundness of the method.
> > >
> > > My primary concern regarding the disconnect between theory and practice stands. In your response to W3, you explicitly acknowledged that the clustering method "works differently" from the pivot selection algorithm used in your theoretical derivations. This confirms that the theoretical guarantees provided in Theorems 1 and 2 (based on CUR decomposition and specific pivot selection assumptions) do not mathematically apply to the actual K-Means implementation used in your algorithm. This gap makes the theoretical section appear disconnected from the proposed method, serving more as a loose inspiration rather than a rigorous proof of the algorithm's effectiveness.
> > >
> > > Furthermore, I am still concerned about the practical trade-off. While ID-LoRA reduces the number of trainable parameters, it introduces significant engineering complexity—specifically the pre-processing requirement and the inability to seamlessly merge weights for inference due to the router mechanism. In practical large-scale deployment, the benefit of saving a few megabytes of memory is often outweighed by the operational overhead and the loss of the "merge-and-deploy" simplicity found in standard LoRA or DoRA. Due to the unresolved theoretical inconsistency and the concerns about practical utility, I will be maintaining my current score.

---

> > > > ### Author Response · Authors · 2025-11-28
> > > >
> > > > **Response to the Question on the Connection Between Theory and Practice:**
> > > >
> > > > We appreciate the reviewer's focus on the link between our theory and implementation.
> > > > It is crucial to clarify that our design is rooted in our theoretical findings, not merely loosely inspired by them.
> > > >
> > > > Specifically, while pivot selection addresses the core challenge in CUR decomposition, we demonstrate that K-Means clustering is a provably superior strategy for our multi-task learning scenario. Our Theorems 1 and 2 establish that this cluster-aware pivot selection yields a tighter error bound than the conventional global pivot selection.
> > > > This insight motivated our use of cluster centroids, which serve as an effective alternative for pivot selection.
> > > >
> > > > As demonstrated in Table 3, our cluster-based method consistently outperforms the global approach, which is consistent with the theoretical predictions of Theorems 1 and 2.
> > > >
> > > > **Response to the Concern on Practical Trade-offs and Deployment Complexity.**
> > > >
> > > > While the inability to merge adapter weights introduces additional computational overhead compared to standard LoRA from a single-task perspective, we wish to clarify that this design represents a necessary trade-off to enable more efficient multi-task serving within a single model.
> > > >
> > > > Standard LoRA's "merge-and-deploy" scheme suits single tasks well but does not scale to multi-task scenarios, which is confirmed by the data in Table 2.
> > > > Its "semi-merged" design, with Part A being inherently merged from the pre-trained parameters, enables not only superior multi-task performance but also a more favorable performance-efficiency trade-off.
> > > >
> > > > The ID-LoRA framework overcomes this limitation through its "semi-merged" design. By keeping Component A directly integrated with pre-trained weights, it achieves both superior multi-task performance and better computational efficiency.
> > > >
> > > > As shown in Figure 3, ID-LoRA exhibits significantly lower inference latency than other non-mergeable multi-task adapters like MoELoRA and HydraLoRA, while nearly matching the speed of merged LoRA/DoRA with only a 0.5% overhead.
> > > > Furthermore, what is perceived as "engineering complexity" is, in fact, the core innovation that enables ID-LoRA to move beyond the single-task paradigm.

---

### Official Review · Reviewer_ztga · 2025-11-03

**Soundness:** 2
**Presentation:** 2
**Contribution:** 2
**Rating:** 4
**Confidence:** 4

**Summary:**

In this paper, the author proposed the ID-LoRA method, which aims to reduce the trainable parameters during efficient fine-tuning. On five diverse benchmarks, the author validates the effectiveness of ID-LoRA on both single- and multi-task scenarios.

**Strengths:**

-	Section 4.3 provides a theoretical foundation for the proposed method, which is interesting.
-	With fewer trainable parameters, ID-LoRA achieves better results in both single-task and multi-task scenarios.

**Weaknesses:**

-	The reasoning behind using MID for initializing matrices $A_i$ is unclear. Please compare/justify against alternatives like SVD or random initialization.
-	The writing is messy and unclear.
    -    Numerous abbreviations (e.g., PMRC, RB, Pivot) are used without definition, hindering comprehension. And the Equation numbering is inconsistent; some have equation numbers, some do not.
    - The terms "good pivot" and "bad pivot" are undefined, making Assumption 3 resemble a definition rather than a theoretical premise. And the max/min inequality relationship needs explanation.
    - In Assumption 2, why is it $A_iB^T$ rather than $BA_i$
-	Need additional ablation studies.
    - Main results are based on Llama-3-8B, but ablation studies use Llama-3.2-3B. This is weird. Ablations should be replicated on Llama-3-8B for consistency.
    -  In Table 3, besides PMRC/SS/RS, include comparisons with non-MID initializations (e.g., random or SVD).
    - In Table 3, why does "w/o RB" perform worse than "w/ RB"? This is counterintuitive since "w/ RB" has more parameters, and "w/o RB" (equivalent to duplicated B matrices) is a subset of the "w/ RB" solution.
    - Need to compare with other efficient LoRA methods (e.g., LoRA-XS, VeRA)
-	Why do some benchmark metrics in Figure 4 degrade as trainable parameters increase?
-	There is a gap between the theory in Section 4.3 and ID-LoRA's practice. The theory approximates optimal matrices $W_i$, but ID-LoRA decomposes the pretrained weight $W$. This implicitly assumes $W$ contains $W_i$, which is questionable: How can pretrained weights encompass downstream task-specific optima? And why can MID approximate this?

**Questions:**

See above

---

> ### Author Response · Authors · 2025-11-20
>
> We sincerely appreciate your thoughtful and constructive comments. We have carefully considered each of your questions and provide detailed responses below.
>
> **W1: Regarding the initialization methods**
>
> Your question about the motivation for MID initialization is important. We did consider alternative approaches such as SVD and random initialization. We finally selected MID for the following core reasons: (1) MID vs. SVD: MID enables the reuse of multiple groups of original parameters, leading to better multi-task generalization; (2) MID vs. Random Initialization: By leveraging the original parameter information, MID more reliably enhances the model's reasoning performance.
>
> As you suggested, we have conducted experiments with SVD and random initialization in multi-task scenarios. The experimental results are as follows:
>
> | Method | MATH |  MMLU | CQA |CODE Pass@1|CODE Pass@5|CODE Pass@10|
> | :---:  | :---: | :---: | :---: | :---: | :---: | :---: |
> | *LLaMA-3-8B*  | | | | | |
> | Randominit  |  **60.2**  |  49.2 |  44.9  |   40.3 |  53.4  |   55.2 |
> | SVDinit  | 47.9 | 35.2 | 44.6 | 33.7 | 42.3 | 45.3 |
> | **ID-LoRA（ours）**  | 59.6 | **51.0** | **47.4** | **41.7** | **54.5** |**58.5**|
>
> The experimental results in the table above also validate our analysis, demonstrating that MID initialization achieves better multi-task modeling capability compared to SVD initialization and random initialization. We will add this experiment along with the analysis to the next version of the paper.
>
> **W2: Clarification and revision of writing issues**
>
> 1. Regarding the issues of undefined abbreviations, we have conducted a review and will provide clear definitions in the next version of the paper:
> (a) Parameter Matrix Row Clustering (PMRC): A K-Means clustering method applied to the row vectors of the parameter matrix.
> (b) Rank Boosting (RB): A technique to enhance the rank of parameter fine-tuning without increasing the number of trainable parameters in the model.
> (c) Pivot: This is not an abbreviation but a specialized term in the context of matrix interpolative decomposition, referring to the key rows or columns selected to form the skeleton matrix.
> In our paper, we only numbered equations that were referenced. We acknowledge that numbering all equations would facilitate easier reading and indexing.
>
> 2. Regarding Assumption 3, "good pivots" refer to sets of pivots that can span the dominant row/column subspace of the original matrix $W$. "Bad pivots" refer to sets that fail to effectively capture this principal subspace. The inequality intuitively reflects the strong dependence of CUR decomposition performance on pivot selection. We will revise it in the next version of the paper.
>
> 3. Regarding the matrix multiplication in Assumption 2, to maintain consistency, we have uniformly revised it to $BA_i$ in the corrected version.

---

> ### Author Response · Authors · 2025-11-20
>
> **W3: Regarding the addition of ablation experiments**
>
> 1. Thank you for your suggestion regarding the need for ablation experiment on the LLaMA3-8B model. We have included the experiment:
>
> | Method | Params(%) | MATH |  MMLU | CQA |CODE Pass@1|CODE Pass@5|CODE Pass@10|
> | :---: | :---: | :---: | :---: | :---: | :---: | :---: | :---: |
> | *LLaMA-3-8B* | | | | | | |
> | LoRA | 1.03% | **60.8** | 48.1 | 46.9 | 39.0 | 48.8 | 51.9 |
> | ID-LoRA(w/o RB) | 0.60% | 59.4 | 50.2 | 46.4 | 40.2 | 50.8 | 53.7 |
> | ID-LoRA(SS+RB) | 0.56% | 58.8 | 50.7 | **47.7** | 40.7 | 53.5 | 56.2 |
> | ID-LoRA(RS+RB) | 0.56% | 59.2 | **51.0** | 47.2 | 40.1 | 52.8 | 55.4 |
> | **ID-LoRA(PMRC+RB)** | **0.56%** | 59.6 | **51.0** | 47.4 | **41.7** | **54.5** |**58.5**|
>
> The results we obtained from the LLaMA3-8B model are consistent with those from the LLaMA3.2-3B model, with both the RB and PMRC modules showing certain performance improvements.
>
> 2. Discussions and experiments regarding the comparison with SVD and Random initialization methods can be found in **W1**.
>
> 3. Our experimental section did not clearly state the parameters for the RB ablation study. To validate the effectiveness of the RB design, we conducted experiments while ensuring that the number of trainable parameters remained the same. Since the method without RB models a smaller rank compared to the method with RB, under the condition of equivalent trainable parameters, the method with RB performs better. The specific parameter quantities and the relationship with ranks in Table 3 are shown as follow:
>
> | Method | #Params(%) | rank |
> | :---: | :---: | :---: |
> |ID-LoRA(PMRC+RB)|0.83%|128|
> |ID-LoRA(w/o RB)|0.77%|32|
>
> 4. Our method exhibits significant differences in the main technical approach compared to efficient LoRA methods such as LoRA-XS and VeRA. While LoRA-XS and VeRA achieve a hundredfold parameter advantage by freezing most of the LoRA modules, our method focuses on multi-task scenarios, simultaneously achieving a certain level of parameter efficiency to realize a trade-off between rank and parameters.
>
> Following your suggestion, we have added comparative experiments with LoRA-XS and VeRA under multi-task settings.
>
> | Method | Params(%) | MATH |  MMLU | CQA |CODE Pass@1|CODE Pass@5|CODE Pass@10|
> | :---: | :---: | :---: | :---: | :---: | :---: | :---: | :---: |
> | *LLaMA-3-8B* | | | | | | |
> | LoRA-XS（r=32） | 0.0028% | 54.6 | 44.7 | 37.7 | 31.9 | 45.5 | 48.9 |
> | LoRA-XS （r=256）| 0.1684% | 56.7 | 46.8 | 39.9 | 35.5 | 47.1 | 50.8 |
> | VeRA （r=32）| 0.0172% | 55.2 | 46.0 | 40.1 | 34.8 | 46.8 | 50.6 |
> | VeRA （r=256）| 0.0178% | 55.9 | 46.2 | 40.7 | 34.6 | 46.4 | 50.3 |
> | **ID-LoRA（r=128）** | **0.56%** | 59.6 | **51.0** | **47.4** | **41.7** | **54.5** |**58.5**|
>
> The experiments demonstrate that our method outperforms LoRA-XS and VeRA in multi-task scenarios. Moreover, the total parameter count of these two methods grows very slowly as the rank increases, while the number of frozen parameters increases significantly, occupying additional GPU memory. This poses a bottleneck in complex multi-task scenarios, making it difficult to achieve a balance between efficiency and effectiveness.
>
> **W4: Regarding the issue in Figure 4**
>
> Due to the heterogeneity among multi-task mixed datasets, conflicts may arise during the learning process, leading to performance degradation. Additionally, since the data shuffle seeds were consistent across different baselines in the experiments, LoRA and ID-LoRA exhibit similar trends in benchmark metrics.
>
> **W5: Clarification on the gap between theory and experiment**
>
> ID-LoRA does not directly decompose the pre-trained weights. Instead, it applies the concept of interpolation decomposition to the matrix $W_i$ for downstream tasks. The core idea is to select key rows(or columns) from the pre-trained weights through clustering and a Router mechanism, followed by linear combination to approximate the key components of $W_i$. These are then multiplied by a shared trainable matrix $B$ to approximate the optimal solution of the matrix $W_i$ required for the downstream task. This approach implicitly assumes that the pre-trained weights contain certain knowledge relevant to the downstream tasks.
>
> If you have any further comment, please feel free to let us know and we are more than glad to discuss with you.

---

### Official Review · Reviewer_Gd5V · 2025-11-03

**Soundness:** 2
**Presentation:** 3
**Contribution:** 2
**Rating:** 4
**Confidence:** 4

**Summary:**

This paper introduces ID-LoRA, a new PEFT framework designed to improve upon LoRA. The approach, inspired by Matrix Interpolative Decomposition, allows the model to achieve a high effective rank while reducing the number of trainable parameters. The authors evaluate ID-LoRA on a diverse set of benchmarks (math, code, MMLU, etc.) and show that it outperforms LoRA and its variants (like DoRA) while using up to 46% fewer trainable parameters.

**Strengths:**

- The rank-vs-parameter trade-off is a well-known limitation of LoRA.
- The method demonstrates clear and consistent performance gains.
- The authors provide a theoretical analysis (Theorems 1 & 2) to back their method.

**Weaknesses:**

- The method is inherently more complex to implement than vanilla LoRA. The cost of the pre-processing step is not discussed.
- The paper does not compare ID-LoRA against LoRI in either single-task or multi-task settings, even though several of the result tables are structurally similar to those in LoRI. Since LoRI is more parameter-efficient than ID-LoRA, including this comparison would provide a clearer picture of the trade-offs.
- The method introduces a new and important hyperparameter, $k$ (the number of clusters). The ablation in the appendix (Table 8) shows performance is sensitive to this.
- The authors rightly admit that the performance gains on mathematical reasoning (GSM8K) are "less pronounced". This suggests ID-LoRA may not be a universal improvement for all task types.
- The RB component feels somewhat separate from interpolative decomposition.
- The experiments are conducted on LLaMA-3-8B and Mistral-7B, which are models from 2023 and 2024. Evaluating the method on more current base models would be necessary to demonstrate its relevance for a 2026 conference.

**Questions:**

- What is the practical, one-time cost (in terms of time and compute) of running k-means clustering?
- The parameter savings seem to come from two places: (1) reusing W, sharing B and (2) the RB technique. Could you provide a clearer parameter breakdown in Table 3?

---

> ### Author Response · Authors · 2025-11-20
>
> We sincerely appreciate your thoughtful and constructive comments. We have carefully considered each of your questions and provide detailed responses below.
>
> As you recommended the use of more current base models, we have performed the supplementary experiments using Qwen3-8B (2025).
>
> ### **W1 The cost of the pre-processing step**
>
> Clustering Preprocessing Cost: With Qwen3-8B model, we employed the k-means-constrained algorithm on a single core of an Intel(R) Xeon(R) CPU E5-2683 v3. Clustering the 7 linear layers in one decoder layer takes approximately 8 minutes. The most computationally intensive layers, namely the up_proj and gate_proj, each account for about 2.5 minutes. Since the clustering of different layers can be performed concurrently, the total time can be reduced to 2.5 minutes in a multi-process scenario. Furthermore, this duration is expected to decrease further with GPU-accelerated optimizations for parallel clustering of individual layers.
>
> Although ID-LoRA introduces an additional step compared to standard LoRA, the clustering process is not data-driven. It needs to be performed only once for a given model and target rank, and does not affect training across different datasets.
>
> ### **W2 Comparison with LoRI**
>
> Although LoRI is a multi-task LoRA variant, it follows a distinct technical path: it involves separate per-task training and adapter merging, unlike ID-LoRA's single-stage training on mixed data with a router. We chose MoELoRA and HydraLoRA as our initial baselines because their methods are similar to ours.
>
> As you suggested, we have added a comparison with LoRI-D to our work on Qwen3-8B, using the same mixed-data training method. The results are presented below:
>
> |Method|#Params(%)|MATH|MMLU|CQA|CODE Pass@1|CODE Pass@5|CODE Pass@10|
> |:---:|:---:|:---:|:---:|:---:|:---:|:---:|:---:|
> |*Qwen-3-8B*|||||||
> |LoRI-D|0.54%|81.9|44.7 |72.8|63.6|75.7|78.2|
> |**ID-LoRA (Ours)**|0.56%|**82.7**|**48.5**|**78.8**|**66.1**|**77.2**|**79.6**|
>
> As the results show, ID-LoRA exhibits clear superiority over LoRI-D in all four tasks. This is because ID-LoRA employs a trainable router to learn expert combinations, whereas LoRI depends on a linear combination governed by manually defined hyperparameters.
>
>
> ### **W3 Hyperparameter $k$**
>
> The hyperparameter $k$ exhibits a certain degree of sensitivity. The ablation study (Table 8) in the paper was designed to explore and select an appropriate value for $k$.
>
> While $k$ is indeed sensitive, its viable range of values is limited. Table 8 shows that when $k$ becomes sufficiently large, performance consistently degrades (we attribute this to the decreasing distinctiveness among the parameter-clustered matrices $A_i$ as $k$ increases). Based on the experimental results in Table 8, we selected $k$ = 4 or 6 for the multi-task experiments.
>
> ### **W4 Mathematical Reasoning**
>
> While the performance improvement of ID-LoRA on mathematical reasoning tasks is indeed 'relatively limited,' in mixed-data training scenarios, it maintains comparable performance to other methods on mathematical tasks while achieving improvements across all other tasks. This itself demonstrates the effectiveness of the ID-LoRA method. Moreover, improvements in mathematical tasks cannot solely rely on supervised fine-tuning through reparameterization and may require specialized learning algorithms specifically designed to enhance mathematical reasoning.
>
> ### **W5 RB component**
>
> The RB component enhances the rank of the ID-LoRA module by reusing the B matrix for activation vectors, thereby improving the representational capacity when handling complex multi-task data. This aligns with ID-LoRA's architecture, which combines clustered interpolative decomposition with a routing mechanism to adapt to complex multi-task scenarios.
>
> The RB component reduces the number of trainable parameters under the same rank setting, which is consistent with the structural effect of parameter freezing after interpolative decomposition in ID-LoRA.
>
> We have added this explanation to the latest version of the Introduction.
>
> ### **W6 Current base model**
>
> We have incorporated the newly released Qwen3-8B model (2025) as the base model and conducted comparative experiments with various efficient fine-tuning methods. The results are presented in the table below:
>
> |Method|#Params(%)|MATH|MMLU|CQA|CODE Pass@1|CODE Pass@5|CODE Pass@10|
> |:---:|:---:|:---:|:---:|:---:|:---:|:---:|:---:|
> |*Qwen-3-8B*|||||||
> |LoRA|1.05%|**84.6**|45.4|67.2|62.9|73.5|77.0|
> |**ID-LoRA (Ours)**|**0.56%**|82.7|**48.5**|**78.8**|**66.1**|**77.2**|**79.6**|
>
> The experimental results in the above table demonstrate that the ID-LoRA method achieves higher accuracy on most tasks, validating that the ID-LoRA architecture remains effective on newly released models.

---

> ### Author Response · Authors · 2025-11-20
>
> ### **Q1 The cost of k-means clustering**
> See the answer in **W1**
>
> ### **Q2 Parameter breakdown**
>
> The parameter reduction indeed stems from these two aspects. We have added the following table detailing the analysis of trainable parameter savings:
>
> |Method|#Params(%)|
> |:---:|:---:|
> |lora|100%|
> |reusing W (Frozen A)|50%|
> |reusing W + RB|12.5%|
>
> Using LoRA as the baseline, reusing W reduces trainable parameters by half. When combined with RB, the trainable parameters are further reduced to 12.5% of LoRA's.
>
> It should be noted that the purpose of Table 3 is not to demonstrate the reduction in trainable parameters, but rather to evaluate whether different technical modules (i.e., reusing W and RB) remain effective and their corresponding performance improvements when the number of trainable parameters is kept approximately constant.
>
> Regarding the information in Table 3, we have added the following table to analyze the improvement in rank achieved by the RB method under an equivalent parameter budget.
>
> |Method|#Params(%)|rank|
> |:---:|:---:|:---:|
> |ID-LoRA(PMRC+RB)|0.83%|128|
> |ID-LoRA(w/o RB)|0.77%|32|
>
> If you have any further comment, please feel free to let us know and we are more than glad to discuss with you.

---

### Author Response · Authors · 2025-12-03

Dear Area Chair,

We would like to express our sincere gratitude for you and reviewers for the constructive review process for our submission “ID-LoRA: Efficient Low-Rank Adaptation Inspired by Matrix Interpolative Decomposition” (Submission ID: 10557).
We sincerely appreciate the time and effort all reviewers have dedicated to evaluating our work. Below, we provide a summary of our responses and revisions based on their valuable feedback.

We have carefully considered every comment from the reviewers and has taken substantial steps to address their concerns, including:

**1. Comprehensive Experimental Expansion:**

(1) We conducted new experiments using the recently released Qwen3-8B (2025) model, as recommended by Reviewer **Gd5V**, to ensure the relevance and applicability of ID-LoRA on contemporary base models.

(2) Additional comparisons with LoRI-D, and VeRA, and other parameter-efficient baselines were performed to provide a more complete picture of ID-LoRA’s advantages.

**2. Enhanced Methodological Clarity:**

(1) We revised ambiguous formulations, clarified mathematical notations (e.g., Equation 4), and provided definitions for all abbreviations (PMRC, RB, etc.) to improve readability and reproducibility.

(2) The theoretical motivation behind using Matrix Interpolative Decomposition (MID) versus alternatives (SVD, random initialization) is now supported by comparative experiments and analysis.

**3. Deeper Ablation and Sensitivity Analysis:**

(1) We extended ablation studies to LLaMA-3-8B for consistency and added analyses on hyperparameter sensitivity (e.g., cluster count $k$), training cost, and inference-time behavior.

(2) Detailed breakdowns of parameter savings and the role of the Rank Boosting (RB) component were provided to clarify the source of efficiency gains.

**4. Addressing Practical Concerns:**

(1) We quantified the one-time preprocessing cost of k-means clustering (approx. 2.5 minutes per layer with parallelization) and discussed the trade-offs between parameter reduction and deployment complexity.

(2) Experiments demonstrate that the inference latency and memory overhead consistently remain the lowest compared to other multi-task efficient fine-tuning methods. After parameter merging, the latency increase is only approximately 0.5% relative to the commonly used LoRA method, while multi-task modeling performance is significantly improved, ensuring the practical usability of ID-LoRA.

**5. Strengthening Theoretical-Practical Alignment:**

 We note that Reviewer **7pCV** raised concerns regarding the connection between theory and implementation. We respectfully clarify that our adoption of k-means clustering constitutes a principled instantiation of the theoretical pivot-selection framework, specifically tailored for multi-task learning scenarios. The experimental superiority demonstrated by our cluster-aware approach (as shown in Table 3) aligns directly with the theoretical guarantees, thereby further substantiating the robustness of our method.

In light of the significant revisions and additional evidence provided, we believe that our paper presents a more robust, comprehensive, and compelling contribution.
We propose ID-LoRA, a semi-merged adapter framework designed for multi-task modeling scenarios. By leveraging a clustering-based mechanism to reuse pretrained weights, it reduces trainable parameters by up to 46% and surpasses mainstream methods such as LoRA, DoRA, and MoELoRA across various benchmarks including mathematical reasoning, code generation, and multi-task learning. Supported by rigorous theoretical analysis and extensively validated through experiments on models of up to 8B parameters, the proposed method achieves performance improvements while maintaining low inference overhead.

Thank you again for your time and consideration. We remain available for any further clarification or discussion.

Sincerely,
The Authors of Submission #10557

---

### Meta-Review · Area_Chair_3oxN · 2026-01-02

**Summary:**

The decision is primarily informed by concerns regarding the theoretical consistency and practical applicability of ID-LoRA. While the method shows potential in reducing trainable parameters and improving efficiency across various benchmarks, its reliance on the MID initialization and the lack of clear theoretical guarantees hinder its overall impact. Additionally, issues with clarity, unexplained hyperparameters, and the introduction of computational overhead make it difficult to assess the real-world utility of the approach.

**Reviewer Concerns:**

The rebuttal provides additional experimental results and clarifies the computational cost of preprocessing. However, the connection between the theoretical analysis and the practical implementation of clustering-based pivot selection remains unclear. While the experimental results are solid, there is a lack of in-depth exploration of hyperparameter sensitivity and performance across larger models. Additionally, the method's real-world applicability is still uncertain due to the engineering complexity introduced by the router mechanism. Without clearer comparisons to other efficient methods and further validation of its practical utility, the paper's contributions remain somewhat unclear.

**Reviewer Scores:**

Reviewers Gd5V and ztga would likely maintain their score. Reviewer 7pCV would maintain their score as the unresolved theoretical inconsistencies and concerns about the practical utility of the method. Reviewer jCmp would likely maintain their score as lack of intuitive clarity remains insufficiently remediated.

---

### Decision · Program_Chairs · 2026-01-26

Reject